# Effect of honey and insulin treatment on oxidative stress and nerve conduction in an experimental model of diabetic neuropathy Wistar rats

**Allampalli Sirisha**[1], **Girwar Singh Gaur**[1]\*, **Pravati Pal**[1], **Zachariah Bobby**[2], **Bharathi Balakumar**[1], **Gopal Krushna Pal**[1]

**1** Department of Physiology, Jawaharlal Institute of Post-graduate Medical Education and Research (JIPMER), Puducherry, India, **2** Department of Biochemistry, Jawaharlal Institute of Post-graduate Medical Education and Research (JIPMER), Puducherry, India

\* drgsgaur@yahoo.com

**Data Availability Statement:** All relevant data are within the manuscript.

## Abstract

Diabetic neuropathy is the most common complication affecting more than 50% of patients with longstanding diabetes. Till date, there are no reports to explain the scientific basis of alternative medicine as an adjunct therapy for treating diabetic neuropathy. Hence, we studied the effect of honey and insulin treatment on hyperglycemia, dyslipidemia, oxidant and anti-oxidant status and nerve conduction in experimental diabetic neuropathy Wistar rats. In this experimental study, forty healthy male Wistar albino rats of 10–12 weeks age, weighing between 150 to 200g were obtained from our institute central animal house. After acclimatization, the rats were divided into control (n = 8) and experimental (n = 32) groups randomly. In the experimental group, type 2 diabetic neuropathy was induced with high fat and high sugar diet for 8 weeks followed by streptozotocin at a dose of 35 mg/kg body weight. Three days after streptozotocin injection, blood glucose levels of rats were measured from fasting samples to confirm diabetes. After the development of diabetes, rats were given standard rodent chow and allowed four more weeks to remain diabetic and to develop neuropathy. Every second week, nerve conduction study was done to confirm neuropathy. All the diabetic rats of experimental group developed neuropathy after 4 weeks of developing diabetes, which was confirmed by significant reduction in conduction velocity of sensory and motor nerve when compared to non-diabetic control group. After the development of neuropathy, these rats were randomly divided into diabetic neuropathy with no treatment group (n = 8) and three treatment groups (n = 8, each). The rats of treatment group were administered with either honey or insulin or honey+insulin for six weeks. After six-weeks of intervention, there was significant decrease in blood glucose and lipids in honey, insulin and honey +insulin treated neuropathy rats, when compared with no treatment group. Malondialdehyde was reduced and total anti-oxidant status improved in all the three treatment groups. There was no significant increase in conduction velocity of sciatic tibial motor nerve in treatment groups when compared with no treatment group. However, the sensory nerve conduction velocity improved significantly in honey+insulin treated neuropathy rats. In conclusion, six-

**Funding:** Our institute (JIPMER) provided intramural fund for conducting PhD thesis as part of protocol. The funders had no role in study design, data collection and analysis, decision to publish, or preparation of the manuscript.

**Competing interests:** The authors have declared that no competing interests exist.

week honey treatment helped in reducing dyslipidemia and oxidative stress. Honey given along with insulin for six-weeks improved sensory nerve conduction velocity in experimental diabetic neuropathy Wistar rats.

## Introduction

Diabetes mellitus is the most common non-communicable disorder worldwide, prevalence of which is increasing steadily. According to international diabetes federation (IDF), India is in second place among the countries with largest numbers of adults with diabetes in 2019; and there are 77.0 million people suffering from diabetes in India [1]. Neuropathy is the most common complication associated with diabetes, which affects about 10% of newly diagnosed patients and more than 50% of patients with longstanding diabetes [2]. The presentation of diabetic neuropathy (DN) is heterogeneous, affecting different parts of the nervous system that expresses with diverse clinical manifestations. Most common among the diabetic neuropathies are chronic sensorimotor distal symmetric polyneuropathy and the autonomic neuropathies [2, 3]. Decreased nerve conduction velocity (NCV), which is frequently subclinical, is considered to be the first objective quantitative indicator of polyneuropathy [3].

Persistent hyperglycemia has been reported to be the major reason for microvascular complications in diabetic neuropathy [4]. The major metabolic changes caused by persistent hyperglycemia are increased polyol pathway flux, elevated oxygen free radical formation, and advanced glycosylation [4]. Increased glucose concentration causes metabolic abnormalities like increased oxidative stress and lipotoxicity, which in turn leads to the endothelial abnormalities, causing decreased nerve blood flow and oxygen supply that results in nerve degeneration [5]. As many of the pharmacological treatments show side effects, complementary therapies are evolved to treat diabetes. Dietary supplements have been investigated by many researchers for preventing or treating type 2 diabetes and its complications [6–8]. As several metabolic risks together leads to neuropathy in diabetes, the dietary supplement that acts on most of these metabolic risks is the best choice for the complementary treatment of DN. Honey is one of the natural supplements which is reported to show protective effects on several metabolic risk factors of diabetes [9–11]. However, till date, there are no reports to explain the scientific basis of honey as an adjunct therapy to the treatment of DN.

Honey is a natural substance with various medicinal properties, which include antibacterial, antihypertensive, hepatoprotective, hypoglycemic and antioxidant effects [9–11]. Studies have shown that honey exerts a hypoglycemic effect and ameliorates oxidative stress in streptozotocin-induced diabetic rats [10, 12]. It has been documented that antioxidant treatment prevents or slows the development of neuropathy in animal models of diabetes, signifying a major contribution of reactive oxygen species in the pathophysiology of DN [13, 14]. Some researchers have studied the effect of honey compared to the anti-diabetic drugs such as metformin and glibenclamide [15, 16]. These studies have reported that when honey was given with metformin or glibenclamide, there was reduction in hyperglycemia and oxidative stress when compared to metformin or glibenclamide alone. Diabetic patients tend to crave for sweet tasting foods as it is usually advised to reduce or omit sugar from their diet. Honey has anti-hyperglycemic and anti-inflammatory effect and it is a potent anti-oxidant [11, 12]. Honey being a sweet tasting substance, can be a good supplementation to give the satisfaction and at the same time help to reduce the complications of DN. Since there are established reports on honey compared to antidiabetic drugs like metformin and glibenclamide [15, 16], we have selected

insulin to compare with honey in our study. Also, we have used streptozotocin to develop diabetes in our study. As streptozotocin causes β-cell destruction, we used insulin treatment modality in the present study. To best of our knowledge, effect of honey on nerve conduction velocity in diabetic neuropathy has not been studied. Therefore, in this study, we assessed the effect of honey on hyperglycemia, dyslipidemia, oxidative stress and nerve conduction and compared it with insulin treatment in diabetic neuropathy rat models and their association with the complications of DN.

## Materials and methods

### Animals

In this experimental study, forty healthy male Wistar albino rats of 10–12 weeks age, weighing between 150 to 200g were obtained from the central animal house of Jawaharlal Institute of Post-graduate Medical Education and Research (JIPMER). Approvals of Institute Scientific Advisory Committee and Animal Ethics Committee, JIPMER, Puducherry were obtained. Animals were housed in individual cages with 12/12 h light/dark cycle and food, water *ad libitum*, in air conditioned room with temperature at $25 \pm 2°$ C. The animals were handled in accordance with the Committee for the Purpose of Control and Supervision of Experiments on Animals (CPCSEA) Guidelines for the care and use of animals for experimental purposes. Animals were acclimatized to the animal room condition for one week prior to the experiment at the Animal Research Laboratory in the Department of Physiology, JIPMER, following which the rats were divided into control (n = 8) and experimental (n = 32) groups randomly.

### Induction of diabetic neuropathy

In the experimental group rats, type 2 diabetic neuropathy was induced as described previously by Dang et al [17]. Experimental group rats were given high fat and high sugar diet for 8 weeks followed by they were injected with streptozotocin (intraperitoneally) dissolved in citrate buffer at a dose of 35 mg/kg body weight. Three days after streptozotocin injection, development of diabetes was confirmed by measuring their glucose level in fasting blood samples. Glucose measurement was performed by glucose-oxidase method using an Accu-Chek Performa Glucometer (Roche Diabetes Care, Mannheim, Germany). Rats with blood glucose concentration of 200 mg/dl or more were considered diabetic. According to Dang et al, after the confirmation of diabetes, rats developed neuropathy by the end of 2 weeks. They confirmed neuropathy by mechanical withdrawal threshold and thermal withdrawal latency tests. In our study, after the confirmation of diabetes, every second week we have done nerve conduction study, a standard electrophysiological investigation to confirm the development of neuropathy. All the diabetic rats of experimental group developed neuropathy after 4 weeks of developing diabetes (at 12th week of study), which was confirmed by significant reduction in conduction velocity of sensory and motor nerve when compared to the value of non-diabetic control group rats (Table 1). After developing neuropathy (12th week of study), the DN rats were randomly divided into 4 groups (Fig 1).

1. DN with no treatment group (n = 8)

2. DN+Honey group (n = 8)

3. DN+Insulin group (n = 8)

4. DN+Honey+Insulin group (n = 8)

**Table 1. Comparison of TBW, FBG and nerve conduction study parameters in all the groups at the baseline (before the intervention).**

| Parameters | Normal Control Group | DN with no treatment group | DN+Honey Group | DN+Insulin Group | DN+Honey+Insulin Group |
|---|---|---|---|---|---|
| TBW (g) | 231.37±5.71 | 222.12±4.61 | 223±5.73 | 221.5±5.31 | 219.25±3.91 |
| FBG (mg/dl) | 93.65±2.35 | 420.02±32.08*** | 414.92±32.23*** | 412.70±28.90*** | 421.45±34.50*** |
| **Sciatic-tibial motor nerve conduction** | | | | | |
| Dis. Lat | 0.52±0.04 | 0.87±0.052*** | 0.89±0.02*** | 0.90±0.04*** | 0.90±0.04*** |
| Prox. Lat | 1.24±0.05 | 1.92±0.026*** | 1.89±0.06*** | 1.97±0.06*** | 1.97±0.05*** |
| SMNCV | 52.98±3.89 | 39.26±2.52* | 40.26±2.30* | 40.00±2.58* | 39.65±2.50* |
| **Sural sensory nerve conduction** | | | | | |
| Latency | 0.74±0.08 | 1.85±0.11*** | 1.90±0.02*** | 1.86±0.10*** | 1.91±0.05*** |
| SNCV | 41.53±2.19 | 22.94±1.03*** | 21.59±1.200*** | 23.06±1.23*** | 21.95±1.50*** |

TBW: Total Body weight, FBG: Fasting Blood Glucose; MNCV: Motor Nerve conduction Velocity; SNCV: Sensory Nerve conduction Velocity NSE: Neuron Specific Enolase. Dist. Lat: Distal latency; Prox. Lat.: Proximal Latency; SMNCV: Sciatic motor Nerve conduction Velocity; SNCV: Sural sensory nerve conduction velocity. Values expressed in Mean±SEM; statistical analysis was done by one way ANOVA. The P values less than 0.05 were considered statistically significant. *Comparison with normal control group

*($P<0.05$)

**($P<0.01$)

***($P<0.001$), #Comparison with diabetic neuropathy control group

#($P<0.05$)

##($P<0.01$)

###($P<0.001$).

## Diet and treatment with honey, insulin and honey+insulin

Honey was procured from Green Planet Trust, Theni, Tamil Nadu, India, which was approved by the Food Safety and Standards Association India [FSSAI]. Moreover, the honey was tested for adulteration using water test and ant test [18]. The treatments with honey, insulin and honey+insulin were given from 13th to 18th week. DN+honey group rats were administered honey, once daily at a dose of 0.5 gm/kg body weight, using an oral cannula for a period of 6 weeks. Honey was diluted with distilled water before it was administered. In previous animal studies, different doses of honey were found to be effective. Erejuwa et al., reported that 1g honey per kg body weight was effective for the hypoglycemic and antioxidant effect [12]. However, Azman et al., showed that 0.2g of honey per kg body weight had the neuroprotective effect [19]. In both studies, honey was given for a duration of 4 weeks. Taking both reports

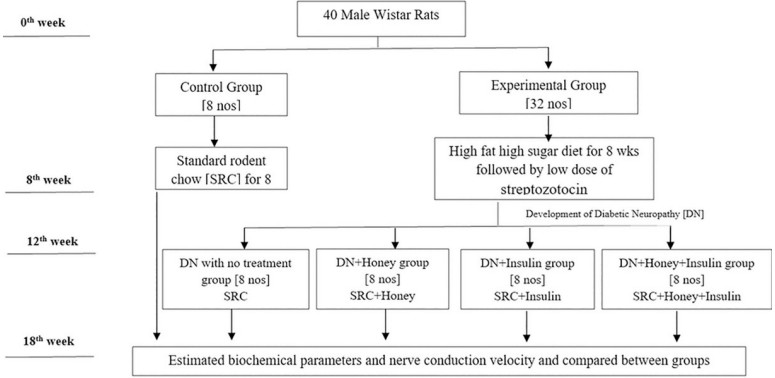

**Fig 1. Schematic representation of the experimental design.**

into consideration, we decided to administer 0.5g of honey per kg body weight for 6 weeks in the present study. Isophane Insulin was obtained from Dept. of Pharmacology, JIPMER. DN+-insulin group rats were injected with isopahane insulin, at a dose of 3 units per day subcutaneously for a period of 6 weeks. DN+honey+insulin group rats received both honey and insulin for a period of 6 weeks (same dose and route of administration as above). During this 6 weeks treatment period, these rats were fed with standard rodent chow. The DN with no treatment group and control group rats were fed with standard rodent chow alone throughout the period of 13[th] to 18[th] week.

Composition of Standard Rodent Chow:

| Composition of SRC | Value (per 1kg chow) |
|---|---|
| Crude Protein | 185.4g |
| Crude Fat | 33.4g |
| Carbohydrates | 650g |
| Crude Fibre | 55g |
| Calcium | 12.2g |
| Phosphorous | 5.8g |
| Total Ash | 56.5g |
| Energy | 3085 kcal/kg |

Two rats died in DN with no treatment group towards the end of 17[th] week. These two rats died unexpectedly without any physical injury. They were active, drinking and eating normally day before their death. These rats were appearing healthy and they did not show any indication of humane end points as per the guidelines of CPCSEA. We did not consider for euthanasia for these two rats as we did not expect their sudden death. We have not allowed them to die without euthanasia, as death happened in these two rats without notice of sickness. In the ethics committee meeting, the mortality aspect of the study protocol was specifically reviewed and approved.

**Recording of nerve conduction study.** Procedures of NCS were performed as per published protocols by Sullivan et al [20] and in compliance with protocols of Diabetic Complications Consortium, formerly the Animal Models of Diabetic Complications Consortium (AMDCC); established by National Institutes of Health. Rats were anesthetized with ketamine (60 mg/kg BW) and xylazine (6 mg/kg BW), injected intraperitoneally. The needle electrodes were cleaned with 70% alcohol, before usage in each animal to maintain pathogen-free status. Recordings were done by using IX-BIO4-SA Small Animal ECG/EMG instrument (iWorx Systems Inc., Dover, New Hampshire, USA).

Sciatic-tibial motor NCV (SMNCV) was determined, by placing the recording electrodes at the dorsum of the foot and stimulating orthodromically with supramaximal stimulation first at the ankle (distal), then at the sciatic notch (proximal). Each time, the distal and proximal latencies were measured from the initial onset of the compound muscle action potential. Then, the difference between distal and proximal latency was calculated. The distance between recording and stimulating electrode was measured and the distance was then divided by the resultant latency to obtain SMNCV. Sural sensory nerve conduction velocity (SNCV) was determined, by recording at the dorsum of the foot and antidromically stimulating with supramaximal stimulation at the ankle. NCV was calculated, by dividing the distance between recording and stimulating electrode by the take-off latency of the sensory nerve action potential.

**Estimation of biochemical parameters.** Two milliliters of blood was collected from the retro-orbital sinus after 6 weeks of treatment, for the estimation of fasting blood glucose (FBG), lipid profile, oxidant and anti-oxidant parameters and neuron specific enolase (NSE).

## FBG, lipid profile, lipid ratios and NSE

The amount of FBG, total cholesterol (TC), triglycerides (TG), LDL- cholesterol (LDL-C) and HDL cholesterol [HDL-C] in serum were estimated, using commercial reagent kits by fully automated clinical chemistry analyzer (ChemWell Awareness Technology, USA). The indices of atherogenic lipid risk factors were assessed, by calculating TC/HDL, TG/HDL, LDL/HDL and Atherogenic index of plasma (AIP) [log10 (TG/HDL-c)], as described previously [21, 22]. NSE was estimated by using ELISA method (Fine Test kit, Wuhan Fine Biotech Co., Ltd, Wuhan, China).

## Oxidative stress parameters

Whole blood reduced glutathione (GSH) was measured by the method described by Beutler et al [23], using 5,5′-dithiobis-(2-nitrobenzoicacid) (DTNB, Ellman's Reagent). The erythro-cyte glutathione peroxidase (GPx) activity was measured, based on the reaction between the glutathione remaining after the action of GPx and 5,5′-dithiobis-(2-nitrobenzoic acid), to form a complex that was absorbed maximally at 412 nm [24]. Lipid peroxidation in plasma was estimated by measuring the levels of malondialdehyde (MDA), a by-product of membrane lipid peroxidation. It was measured according to method described by Yagi et al [25]. The MDA present in the plasma sample reacted with thiobarbitutic acid (TBA) resulting in the formation of coloured compound (MDA-TBA complex), which was measured spectrophoto-metrically at 530nm. Total antioxidant status (TAS) was measured by ferrous-reducing antiox-idant power (FRAP) assay [26]. Though Oxygen Radical Absorbance Capacity (ORAC), Trolox Equivalent Antioxidant Capacity (TEAC) and Diphenyl-β-picrylhydrazyl (DPPH) tests are better tests for measuring TAS, due to the non-availability of the measurement meth-ods in our laboratory, we could not do these tests in the present study. However, reports sug-gests that FRAP assay is comparable with these methods [27, 28] and it is used in various studies for measuring the TAS. Also, FRAP assay has high sensitivity and precision. Taking these points into consideration, we selected FRAP assay in our study for the estimation of TAS.

## Statistical analysis

SPSS version 19 (SPSS Software Inc., Chicago, IL, USA) was used for statistical analysis. All the data were expressed as mean ± SEM. The statistical significance between the groups was analyzed by using one way analysis of variance (1-way ANOVA) with post-hoc Bonferroni analysis. The association of NSE with other parameters was assessed by Spearman correlation analysis. The difference was considered statistically significant, if the value was less than 0.05.

# Results

## Confirmation of diabetic neuropathy

Table 1 shows the data of body weight, FBG and NCS in all the group of rats before starting the intervention (baseline estimation). There was no significant difference in body weight in any of the experimental groups, when compared to control group before intervention. How-ever, FBG was increased significantly in all the experimental group rats, when compared to normal control rats. Also, there was significant reduction of conduction velocity of sensory and motor nerves in the experimental group rats, when compared to normal control group before intervention.

## Effect of six week intervention on TBW and FBG

TBW was significantly less in DN with no treatment group compared to control group, whereas TBW was significantly more in treatment groups when compared with DN with no treatment group. FBG remained significantly high in all groups except DN+honey+insulin group when compared to control group. After six-weeks of intervention, FBG was significantly less in all treatment groups when compared with DN with no treatment group. Also, FBG was significantly less in DN+insulin and DN+honey+insulin treated groups, when compared with DN+honey treated group (Table 2). The pre-post percentage changes in TBW and FBG are depicted in Fig 2. The percentage reduction in FBG was significantly more in DN+honey+-insulin group (p<0.05), when compared to DN+honey group.

## Effect of six week intervention on lipid profile and lipid risk factors

Lipid profile and all lipid risk factors were significantly high with the exception of HDL-C, which was significantly less in DN with no treatment group when compared with control group. Lipid risk factors, TC and LDL-C remained significantly high, whereas HDL-C remained significantly less in DN+honey group when compared to control group. In DN+insulin group, HDL-C was significantly high and LDL-C was significantly less when compared to control group. When compared to DN with no treatment group, TC, LDL-C and all lipid risk factors were significantly less, only in DN+insulin and DN+honey+insulin groups. There was significant decrease in AIP and significant increase in HDL-C, in honey+insulin treated group when compared with honey treated group (Table 2). The pre-post percentage change in lipid profile is depicted in Fig 3. Percentage changes in TC and TG were significantly more in

**Table 2. Comparison of TBW, FBG, lipid profile and lipid risk factors in all the groups after six weeks of intervention.**

| Parameters | Normal Control Group | DN with no treatment group | DN+Honey Group | DN+Insulin Group | DN+Honey+Insulin Group |
|---|---|---|---|---|---|
| TBW (g) | 244.62±5.81 | 204.5±4.66*** | 232.25±4.81## | 232.75±6.10## | 227.12±2.34# |
| FBG (mg/dl) | 98.62±1.59 | 492.66±6.02*** | 187.86±5.88***,### | 121.8±3.80**,###,††† | 106.93±3.21 ###,††† |
| TC (mg/dl) | 105.95±8.92 | 209.10±15.60*** | 155.66±10.29*,# | 149.17±10.90## | 129.39±9.22### |
| TG (mg/dl) | 108.11±9.80 | 204.94±23.25*** | 151.70±18.36 | 130.95±13.05# | 119.47±8.17## |
| HDL-C (mg/dl) | 40.97±1.67 | 20.62±1.31*** | 28.17±1.67***,# | 32.33±1.34**,### | 35.65±1.57###,† |
| VLDL-C (mg/dl) | 21.62±1.96 | 40.98±4.65*** | 30.34±3.67 | 26.19±2.61# | 23.89±1.63## |
| Non-HDL-C | 64.98±9.64 | 188.48±15.73*** | 127.49±10.12**,## | 116.84±11.46*,## | 93.74±9.06### |
| Non HDL-C/HDL-C | 1.63±0.27 | 9.38±1.09*** | 4.65±0.46**,### | 3.70±0.44### | 2.67±0.29### |
| TC/HDL-C | 2.63±0.27 | 10.38±1.09*** | 5.65±0.46**,### | 4.70±0.44### | 3.67±0.29### |
| TG/HDL-C | 2.67±0.27 | 9.92±0.83*** | 5.78±1.15*,## | 4.07±0.40### | 3.38±0.23### |
| AIP | 0.41±0.04 | 0.98±0.03*** | 0.71±0.06***,## | 0.59±0.04### | 0.52±0.03###,† |

DN: Diabetic Neuropathy; FBG: Fasting blood glucose; TC: Total Cholesterol; TG: Triglycerides; HDL-C: High density lipoprotein Cholesterol; LDL-C: Low density lipoprotein Cholesterol; AIP: Atherogenic Index of plasma.

Values expressed in Mean±SEM; The P values less than 0.05 were considered statistically significant. *Comparison with normal control group

*(P<0.05)

**(P<0.01)

***(P<0.001), #Comparison with diabetic neuropathy control group

# (P<0.05)

## (P<0.01)

### (P<0.001), †Comparison with honey treated group

† (P<0.05)

†† (P<0.01)

††† (P<0.001)

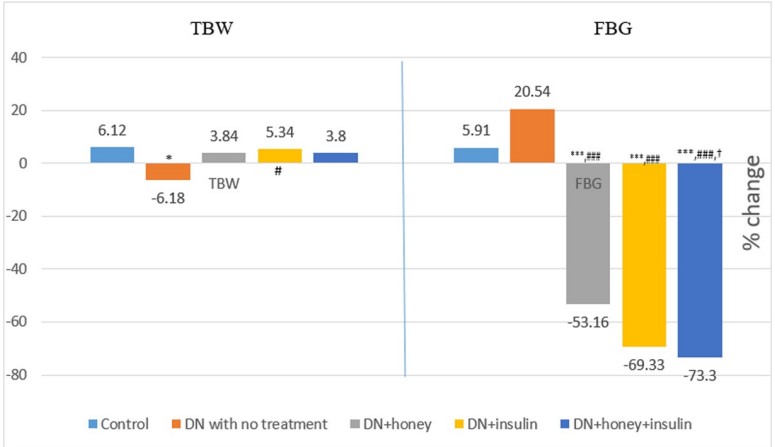

**Fig 2. Percentage change in TBW and FBG before and after six weeks of intervention in all the groups.** TBW: total body weight; FBG: Fasting blood glucose P value less than 0.05 were considered statistically significant. *Comparison with normal control group: *(P<0.05); **(P<0.01); ***(P<0.001), #Comparison with diabetic neuropathy control group:# (P<0.05); ## (P<0.01); ### (P<0.001), †Comparison with honey treated group: † (P<0.05); †† (P<0.01); †††(P<0.001).

DN+honey+insulin group (p<0.05) when compared to DN with no treatment group. Other groups did not show any significant change. However, the percentage changes in HDL and AIP were significantly more in DN+insulin group (p<0.01) and DN+honey+insulin (p<0.001) when compared to DN with no treatment group. HDL showed significantly more percentage change in DN+honey+insulin group (p<0.01) when compared to DN+honey group.

## Effect of six week intervention oxidative stress parameters and NSE

When the post-six week values were compared across the groups, the values of MDA, GPx and NSE were significantly high in DN rats with no treatment when compared with normal healthy control group rats. Also, the values of MDA, GPx and NSE were significantly less and TAS was significantly more in DN+honey, DN+insulin and DN+honey+insulin groups, when

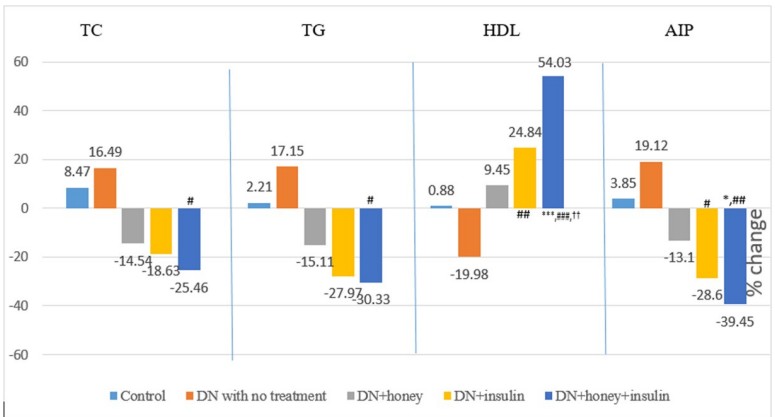

**Fig 3. Percentage change in lipid profile before and after six weeks of intervention in all the groups.** TC: Total cholesterol; TG: Triglycerides; HDL: High density lipoprotein cholesterol; AIP: Atherogenic index of plasma; P value less than 0.05 were considered statistically significant. *Comparison with normal control group: *(P<0.05); **(P<0.01); ***(P<0.001), #Comparison with diabetic neuropathy control group:# (P<0.05); ## (P<0.01); ### (P<0.001), †Comparison with honey treated group: † (P<0.05); †† (P<0.01); ††† (P<0.001).

**Table 3. Comparison of oxidative stress parameters and NSE in all the groups after six weeks of intervention.**

| Parameters | Normal Control Group | DN with no treatment group | DN+Honey Group | DN+Insulin Group | DN+Honey+Insulin Group |
|---|---|---|---|---|---|
| MDA (µmol/l) | 6.38±0.57 | 13.17±1.56*** | 7.48±0.34## | 8.53±1.14# | 6.98±1.00## |
| GPx (u/g Hb) | 15.00±0.82 | 28.90±3.46*** | 15.13±0.96### | 17.07±1.53## | 14.61±2.29### |
| TAS (µmol/l) | 287.09±9.31 | 187.74±4.46*** | 307.92±8.92### | 246.09±7.18*,##,†† | 290.16±13.43###,‡ |
| GSH (mg/g Hb) | 66.99±5.72 | 38.04±4.61** | 56.22±2.03 | 52.02±6.32 | 63.64±5.40# |
| NSE (pg/ml) | 1186.91±56.68 | 1548.28±45.19*** | 1233.02±25.85### | 1288.09±39.20## | 1197.61±34.59### |

DN: Diabetic neuropathy; MDA: Malondialdehyde; GPx: Glutathione Peroxidase; GSH: Reduced Glutathione; TAS: Total anti-oxidant status; NSE: Neuron Specific Enolase.

Values expressed in Mean±SEM; The P values less than 0.05 were considered statistically significant. *Comparison with normal control group

*($P<0.05$)

**($P<0.01$)

***($P<0.001$), # Comparison with diabetic neuropathy control group

# ($P<0.05$)

# # ($P<0.01$)

# # # ($P<0.001$), †Comparison with honey treated group

†($P<0.05$)

††($P<0.01$)

‡ Comparison with insulin treated group: ‡($P<0.05$).

compared with DN with no treatment group. GSH was significantly increased in honey+ insulin treated group alone, when compared with DN with no treatment group. TAS was remained significantly less in DN+insulin group, when compared to DN+honey and DN +honey+insulin group (Table 3). The pre-post percentage change in oxidative stress parameters and NSE are depicted in Figs 4 and 5. TAS showed significantly more percentage change in all the treatment groups when compared to DN with no treatment group. Also, when insulin group was compared to honey group, the percentage change in TAS was significantly less (p<0.05). MDA showed significantly more percentage change in DN+honey and DN+honey +insulin groups (p<0.05), when compared to DN with no treatment group. NSE showed

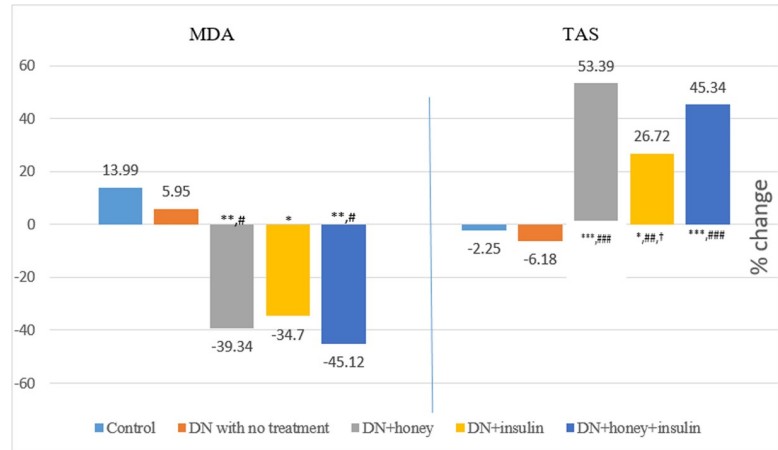

**Fig 4. Percentage change in MDA and TAS before and after six weeks of intervention in all the groups.** MDA: Malondialdehyde; TAS: Total antioxidant status; P value less than 0.05 were considered statistically significant. *Comparison with normal control group: *(P<0.05); **(P<0.01); ***(P<0.001), #Comparison with diabetic neuropathy control group:# (P<0.05); ## (P<0.01); ### (P<0.001), †Comparison with honey treated group: † (P<0.05); †† (P<0.01); ††† (P<0.001).

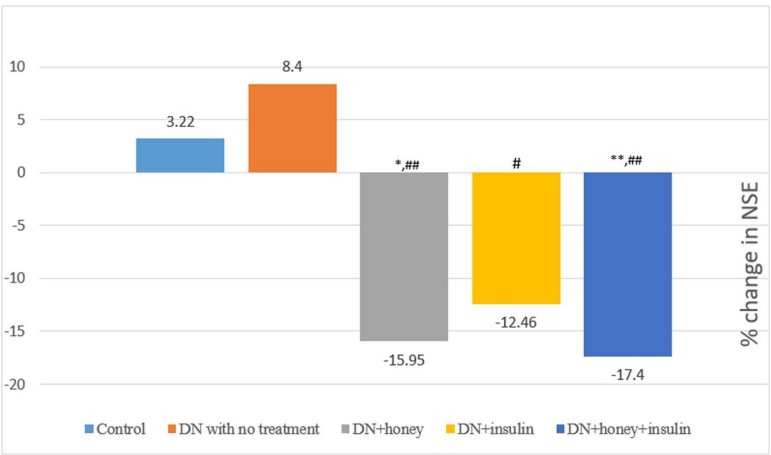

**Fig 5. Percentage change in Neuron specific enolase (NSE) before and after six weeks of intervention in all the groups.** P value less than 0.05 were considered statistically significant. *Comparison with normal control group: *(P<0.05); **(P<0.01); ***(P<0.001), #Comparison with diabetic neuropathy control group:# (P<0.05); ## (P<0.01); ### (P<0.001).

significantly more percentage change in all the treatment groups when compared to DN with no treatment group.

## Effect of six week intervention on nerve conduction

In DN with no treatment group and DN+honey group, there was significant increase in motor and sensory nerve distal latency and significant reduction in SNCV, when compared to control group. In DN+insulin group, there was significantly high sensory latency and low SNCV compared to normal control group. When compared with DN with no treatment group, sciatic-tibial motor nerve latency was significantly less in insulin and honey+insulin treated groups. There was no significant increase in NCV of sciatic tibial motor nerve in all the treatment groups. Sensory nerve latency was significantly less in all the treatment groups when compared with DN with no treatment group. But, the honey+insulin treated group alone showed significantly more NCV of sensory nerve. There was no significant change found between the treatment groups (Table 4).

**Table 4. Comparison of nerve conduction study parameters in all the groups after six weeks of intervention.**

| Parameters | Normal Control Group | DN with no treatment group | DN+Honey Group | DN+Insulin Group | DN+Honey+Insulin Group |
|---|---|---|---|---|---|
| Motor nerve Distal latency | 0.55±0.06 | 0.97±0.04*** | 0.80±0.04** | 0.75±0.04# | 0.71±0.03# |
| MNCV | 50.24±5.22 | 35.43±2.37 | 41.75±1.91 | 44.38±3.59 | 46.48±7.23 |
| Sensory Nerve Latency | 0.80±0.03 | 1.92±0.10*** | 1.50±0.06***,## | 1.42±0.10***,## | 1.33±0.05***,### |
| SNCV | 37.10±2.25 | 21.07±1.91*** | 25.95±1.67** | 28.56±1.89* | 31.45±1.72## |

MNCV: Motor Nerve conduction Velocity; SNCV: Sensory Nerve conduction Velocity.

Values expressed in Mean±SEM; The P values less than 0.05 were considered statistically significant. *Comparison with normal control group

*(P<0.05)

**(P<0.01)

***(P<0.001), #Comparison with diabetic neuropathy control group

#(P<0.05)

##(P<0.01)

###(P<0.001).

## Discussion

In the present study, six-week treatment of honey given with insulin in an experimental model of diabetic neuropathy Wistar rats resulted in improvement of nerve function, which was manifested by reduction in the latency of sensory and motor nerve action potential with near normalization of NSE. It was also accompanied by reduced hyperglycemia, lipotoxicity and oxidative stress.

Ingestion of high fat and high sugar diet for 8 weeks followed by single dose injection of streptozotocin (35 mg, i.p.), was the procedure followed for developing type 2 diabetes in the present study [17]. Similar methods have recently been practiced for effective induction of diabetic neuropathy, an experimental model suitable for pharmacological screening of various preparations [29, 30]. High fat diet causes insulin resistance in peripheral tissues due to lipotoxicity, while low dose of streptozotocin induces mild defect in insulin secretion [30]. High fat diet and low dose streptozotocin model therefore successfully mimics natural progress of diabetes development as well as metabolic features of human type 2 diabetes.

There was reduction in body weight in all the experimental group rats after administration of streptozotocin (Table 1). In streptozotocin induced diabetic rodents, decreased body weight is a commonly observed feature, which is due to wasting of fatty acid and protein stores induced by insulin deficiency [31]. There are controversial reports on the effect of honey on body weight. Some studies reported no change or reduction in body weight after honey treatment [32, 33], whereas others reported that honey increased body weight [9, 12, 34]. In the present study, honey as well as honey given with insulin treatment showed increased body weight (3.84% and 3.8% respectively). The changes in body weight were significantly higher in honey treated group (p<0.01) and in honey+insulin treated group (p<0.05) compared to DN with no treatment group, in whom the body weight decreased (6.18%). The weight gain in the DN rats after honey treatment might be due to the fructose, which is the major component of honey [9, 35].

At the baseline recording, all the experimental groups showed significantly increased FBG when compared to control group (Table 1) indicating the development and maintenance of diabetic stage in all the experimental group rats. Studies reported that honey reduces blood glucose in diabetic rats [10, 12]. Our observation was similar to these reports, as honey reduced blood glucose in DN rats and the reduction was 53.16% from its initial value over a period of 6 weeks (Fig 2). However, the level of blood glucose (Table 2) was still significantly higher (p<0.001) than those of the non-diabetic control rats and in the diabetic range (187.86 mg/dl), which explains that honey exerts a moderate glycemic control. Insulin treatment reduced the FBG value by 69.33%, but it was still higher compared to the non-diabetic control rats (p<0.01) (Table 2); however, it was significantly less (p<0.001) compared to the FBG values in honey treated group. When honey was given with insulin for 6 weeks, the decrease in FBG was more (73.3%), which was comparable to the values in non-diabetic control rats (Table 2). Even though the mechanism of hypoglycemic effect of honey is not clearly understood, some studies reported that fructose component of honey enhances the hepatic glucose uptake and glycogen storage, while also reducing glycemia and increasing insulin levels [9, 10]. Also, honey contains minerals like zinc and copper, which are essential for insulin secretion and glucose metabolism [10, 12].

Independent of glycemic effect, dyslipidemia is one of the major risk factors for development of diabetic neuropathy [36, 37]. Honey reduced TC by 14.54%, TG by 15.11% and AIP by 13.1% and increased HDL-C by 9.45% (Fig 3); and the values were significantly less (TC: p<0.05, AIP:p<0.01) and HDL-C was significantly more (p<0.05) compared to DN rats with no treatment (Table 2). This anti-lipidemic effect of honey is similar to the report of Erejuwa

et al and Alagwu et al [15, 38]. The improvement in lipid profile could be due to the presence of flavonoids in honey, as flavonoids of plant extracts are reported to reduce the cholesterol level [39]. Alagwu et al stated that honey reduces serum levels of cholesterol by increasing its secretion through bile [38]. In the present study, insulin increased HDL-C (24.84%) and reduced other lipid parameters (TC:18.63%, TG:27.97%, AIP:28.6%); and the values were significantly less (TC:$p<0.01$, TG:$p<0.05$, AIP:$p<0.001$) and HDL-C was significantly more ($p<0.001$) compared to DN rats with no treatment. When honey and insulin were given together, they further improved the lipid profile (increased HDL-C by 54.03%, reduced TC by 25.46%, TG by 30.33% and AIP by 39.45%). Significant increase in HDL-C ($p<0.05$) and decrease in AIP ($p<0.005$) was observed in honey+insulin treated DN rats when compared to honey alone treated DN rats (Table 2). Mechanism of action of insulin on amelioration of dyslipidemia is well established [40]. The hypolipidemic effect of honey could be due to the plausible role of insulin, as it has been reported that honey improves serum insulin level in streptozotocin induced diabetic rats [12].

The persistent hyperglycemia and dyslipidemia result in increased release of reactive oxygen species and causes oxidative stress, which finally reduces the blood flow to the nerves and causes nerve degeneration [4, 5]. Several studies have reported the antioxidant effect of honey [9, 11, 12]. In our study, six week treatment of honey reduced the oxidant MDA level (39.34%) compared to 34.7% by insulin. Also, honey treatment improved total antioxidant status (53.39%) compared to 26.72% by insulin in DN rats (Fig 4). Thus, the effect of honey treatment on oxidative stress parameters was found to be better than insulin. Also, TAS value was significantly higher in honey treated DN rats compared to insulin treated DN rats in total plasma concentration ($p<0.01$, Table 3) and in percentage change (Fig 4). This indicates that honey acts mainly through its anti-oxidant mechanism, whereas insulin works mainly through its glycemic control and hypolipidemic effect. The antioxidant effect of honey might be mediated by the ascorbic acid, flavonoids and phenolic components of honey [9, 10], which are established as potent antioxidants [41, 42].

In the present study, there was significant reduction in nerve conduction in experimental group rats when compared to controls (Table 1), indicating the successful development of neuropathy in these rats. Compared to DN rats with no treatment group, sensory nerve latency was significantly less in honey treated group ($p<0.01$) and it was comparable to insulin treated group ($p<0.01$); and the sensory nerve latency was further reduced in honey+insulin treated group ($p<0.001$). In SNCV, insulin did not show any significant change compared to DN with no treatment group, but when insulin was given with honey, SNCV was significantly more ($p<0.01$) (Table 4). This indicates that honey has promotive effect on nerve function and the effect is better seen in sensory neurons. The sciatic-tibial motor nerve conduction velocity did not show significant improvement in the treatment groups. This might be due to the low dose or less duration of the treatment period. As mentioned earlier, the impairment in nerve conduction in neuropathy is due to the persistent hyperglycemia and dyslipidemia, which lead to oxidative stress and reduced blood flow to the nerves causing nerve degeneration. Our study findings indicate that though insulin was able to maintain the glycemic control and ameliorate dyslipidemia, it could not decrease the oxidative stress to the same extent. On the other hand, honey was able to reduce the oxidant level and improve antioxidant status considerably, but its glycemic control and hypolipidemic effect was moderate. When honey was given along with insulin, the sensory nerve conduction improved, which might be due to the combined effect of reduction in glucotoxicity, dyslipidemia and rise in antioxidant level. This was further supported by the reduction in NSE.

NSE is a biochemical marker which can be measured in both blood and cerebrospinal fluid. NSE is an intracellular enzyme principally located in neuronal and glial cells [43]. Due to this

localization, NSE is released into the circulation of cerebrospinal fluid and blood when the nerve is injured [43, 44]. Studies suggest that NSE is a potential biomarker for assessing neuronal damage (neuropathy marker) [43–46]. Li et al reported that NSE is higher in DN patients when compared to normal controls [44]. In our study, after six weeks treatment, we observed significant reduction in NSE in all the treatment groups (p<0.01) compared to DN with no treatment group and this reduction was comparable to the normal control group. This might be an indication of nerve regeneration, as decrease in NSE is a marker of improvement in nerve structure and function [43, 44]. Also, the degree of reduction was better in honey (15.95%) and honey+insulin group (17.4%) compared to insulin alone treated group (12.46%) (Fig 5). NSE was also correlated positively with TC in the honey treated group, and with GPx in honey+insulin treated group showing the possible link between improvement in nerve function and reduction in dyslipidemia and oxidative stress. From the present study findings, it can be stated that since honey has anti-hyperglycemic, hypolipidemic and antioxidant effect, its supplementation along with insulin treatment might help as an adjuvant in treating DN and its complications.

## Limitations of the study

The sample size was only 8 in each group in the study, as animal ethics committee did not permit use of more animals for this work. Therefore, the multiple regression analysis could not be done to assess the individual contribution of various factors determining the effects of honey and insulin therapy on nerve conduction velocity. ORAC, TEAC, DPPH and GSSG tests are better tests for measuring total antioxidant activity and oxidative stress. Due to the non-availability of the measurement methods in our laboratory, we could not do these tests in the present study.

However, conduct of the work in controlled environmental conditions including room temperature and standard prescribed nutrition for all the age and gender matched rats are the merits of the study. Further, effect of six weeks of honey supplementation as an adjunct therapy in diabetic neuropathy rats, which is close to almost four years of treatment of honey in human beings [47, 48] is another novelty of this study. Therefore, the effect of long-term use of honey should be tested in diabetic patients to assess the amelioration of glucotoxic, lipotoxic and neurotoxic effects by honey in diabetic neuropathy.

## Conclusion

In conclusion, six-week honey treatment helped in reducing oxidative stress, dyslipidemia and hyperglycemia. Also, honey given with insulin for six-weeks improved sensory nerve conduction velocity in experimental diabetic neuropathy Wistar rats. Further studies are needed to explore the molecular mechanism of honey for its beneficiary effect in diabetic neuropathy.

## Supporting information

**S1 File.**
(PDF)

## Author Contributions

**Conceptualization:** Allampalli Sirisha, Girwar Singh Gaur, Pravati Pal.

**Data curation:** Allampalli Sirisha, Bharathi Balakumar.

**Formal analysis:** Allampalli Sirisha, Bharathi Balakumar.

**Funding acquisition:** Girwar Singh Gaur.

**Investigation:** Allampalli Sirisha, Bharathi Balakumar.

**Methodology:** Allampalli Sirisha, Pravati Pal.

**Project administration:** Gopal Krushna Pal.

**Resources:** Zachariah Bobby.

**Software:** Allampalli Sirisha.

**Supervision:** Girwar Singh Gaur, Pravati Pal, Zachariah Bobby, Gopal Krushna Pal.

**Validation:** Zachariah Bobby.

**Writing – original draft:** Allampalli Sirisha.

**Writing – review & editing:** Girwar Singh Gaur, Pravati Pal, Zachariah Bobby, Gopal Krushna Pal.

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
