## [Decision Letter · Decision Letter 0]

4 Aug 2020

PONE-D-20-18538

Effect of honey and insulin treatment on oxidative stress and nerve conduction in an experimental model of diabetic neuropathy Wistar rats

PLOS ONE

Dear Dr. Gaur,

Thank you for submitting your manuscript to PLOS ONE. After careful consideration, we feel that it has merit but does not fully meet PLOS ONE’s publication criteria as it currently stands. Therefore, we invite you to submit a revised version of the manuscript that addresses the points raised during the review process.

Please address all the comments and critiques made by the reviewers. Of particular note is: 

The rationale for using honey - why?

The justification for dosage of honey employed?

The model of diabetic neuropathy - how do you prove and validate this?

Several methodological issues that need to be addressed.

Numerous grammatical and formatting errors that should be corrected. 

We look forward to receiving your revised manuscript.

Kind regards,

M. Faadiel Essop

Academic Editor

PLOS ONE

Journal Requirements:

Reviewers' comments:

Reviewer's Responses to Questions

**Comments to the Author**

1. Is the manuscript technically sound, and do the data support the conclusions?

Reviewer #1: Partly

Reviewer #2: Partly

Reviewer #3: Yes

2. Has the statistical analysis been performed appropriately and rigorously? 

Reviewer #1: I Don't Know

Reviewer #2: Yes

Reviewer #3: No

3. Have the authors made all data underlying the findings in their manuscript fully available?

Reviewer #1: Yes

Reviewer #2: No

Reviewer #3: Yes

4. Is the manuscript presented in an intelligible fashion and written in standard English?

Reviewer #1: No

Reviewer #2: Yes

Reviewer #3: No

5. Review Comments to the Author

Reviewer #1: In the present MS, Authors Allampalli Sirisha et collaborators, investigated the impact of honey and insulin treatment on oxidative stress and nerve conduction streptozotocin-induced diabetic rats.

This is an interesting study, however, to my point of view several points hamper its publication:

First sentence of the abstract “Diabetic neuropathy is the most common and debilitating complication affecting more than 50% of patients with longstanding diabetes.” I am not sure the word “debilitating” is appropriated.

Authors wrote “Diabetic neuropathy was induced with high fat and high sugar diet for 8 weeks followed by low dose of streptozotocin”. How can Authors be sure rats developed diabetic neuropathy? What was the criteria? Diabetic status is defined according to fasting hyperglycemia; how specific diabetic neuropathy assessed here?

Lane 64 “Also, malondialdehyde an oxidant” style must be improved.

Lane 69 “improve d”

Lane 146 “dose of 0.5 gm/kg body”

Lane 146 no information is provided concerning administration frequency of honey.

Lane 154 “None of the humane end points”

Lane 160 “Two ml of blood sample was collected”

Lane 162 Authors wrote “the neuropathy marker neuron specific enolase (NSE).” I am not sure NSE is specific of neuropathology. Authors should develop on this.

Lane 175. Authors wrote “Thiobarbituric acid (TBA) was measured by the method of Yagi et al. [21] and expressed in terms of malondialdehyde (MDA).” Actually, TBA test evaluates substances that can react with thiobarbituric acids (TBARS for thiobarbituric acid reactive subtances). Then it can certainly not just be “expressed in terms of MDA”.

Lane 176 Authors wrote “Total antioxidant status (TAS) was measured by ferrous-reducing antioxidant power [FRAP] assay”. I am not sure FRAP is sufficient to evaluate TAS.

Table 1. Three significant numbers are sufficient.

Lane 242 “the percentage change in HDL and AIP was significantly more in both …” A word seems missing in the sentence.

Figures 2 and 3 bring duplicate information of table 1.

Lane 253 “MDA, GPx and NSE were significantly more, whereas…” text should be more specific, more intense? More impacted…? Same remarks for lane 256

Figures 4 and 5 bring duplicate information of table 2.

Same remark for table 2: please note three significant numbers are sufficient. Then table message is more readable than when containing too may numbers.

Lane 289 “significant re duction”

Table 4. How come a r coefficient be superior to 1? For TG, r was equal to 6.6481! Again, three significant numbers are sufficient. I think this table should be removed.

Figure 6. there is a typo “advance glycation end products”. I do not understand why authors put such a schematic scheme. It proposes no hypothetical explanation of honey treatment in their experimental model of DN.

Authors should provide a table with information about regimen in term of composition for standard rodent chow (SRC) and SRC plus honey.

Authors should consider adding a “limitations” section in their discussion.

Reviewer #2: Review on the manuscript entitled: “Effect of honey and insulin treatment on oxidative stress and nerve conduction in an experimental model of diabetic neuropathy Wistar rats” by Sirisha A et al.

In this article the aim is to explain the scientific basis of alternative medicine as an adjuvant therapy in the treatment of diabetic nephropathy, specifically honey in this study. Therefore, they investigated effect of honey with or without insulin on hyperglycemia, dyslipidemia, oxidant and antioxidant status, and nerve conduction in an experimental model of diabetic neuropathy Wistar rats. The authors induced diabetes using a high fat and high sugar diet followed by a low dose of STZ. The initial phase of the study was carried over 8 weeks where the rats received the high fat and high sugar diet followed by a low dose of STZ. Thereafter, the rats received a standard diet for 4 weeks and lastly, a six week period of treatment. The results show that honey+insulin improved sensory nerve conduction velocity and reduced dyslipidemia and oxidative stress.

Overall the study was written relatively well highlighting all the different aspects of the study. Furthermore, the design is made very clear with the schematic diagram and it also made use of one of the gold standard markers/ indicators of neuropathy. Having said that, there are some aspects that the authors should address.

1. The aim of the study was to explain scientific basis of adjunct therapy in the treatment of diabetic neuropathy, in this case, honey. Can the authors perhaps include a statement justifying inclusion of insulin as well?

2. And still on that aspect, their model of diabetes is type 2, so why not go for drugs such as metformin or any other drugs commonly used in the treatment of type 2 diabetes?

3. In the abstract specify the STZ dose used instead of only saying “low dose of STZ” and the type of diabetes as well also in the materials and methods sections as the type only appears in the discussion section.

4. The methodology in the abstract omits the 4 week period where the rats were on the standard diet with the nerve conduction study being carried out. It also does not mention how neuropathy was confirmed.

5. In table 1, please include the baseline weights or measured before treatment to give a clearer picture to the significant weight differences.

6. I find some sentences in the section on “Effect of six week intervention on lipid Profile and lipid risk factors” confusing. For example, the sentence in lines 231 to 232….”Lipid profile and lipid risk factors were significantly high and HDL-C was significantly less…..” Isn’t HDL-C part of the lipid profile? Perhaps using conjunction such as: “with the exception of” will help with this.

7. Also paraphrase the statement in lines 400 to 402 as it stands it is confusing.

8. Please paraphrase for clarity the sentence in lines 288 to 291…”Comparison of NCS…..in DN+insulin group”

9. Can the authors include a reference to the point in lines 426- 427…”almost close to two years of treatment of honey in human beings..”

Minor issue

Line 399 though instead of through

Reviewer #3: The topic under study appears to be an interesting and potentially important one, especially as diabetes is a growing problem globally. It is however, a pity that the authors do not properly motivate why they think honey would be able to restore impaired neuronal conduction in a diabetic model. I think this important even though there are some positive findings in this study. I would also like to say that the manuscript can benefit from language editing as there are several spelling and grammatical issues.

The study is fairly well designed with respect to the model and the types of tests proposed, but there are some problems, as outlined below:

The study appears to be controlled sufficiently through the positive and negative controls employed in this study. It is a pity that some rats died unexpectedly in thus study. It may be advisable to continually measure glucose levels in future studies of this nature, as this is not unheard of in STZ induced diabetic models. One criticism of the intervention would be that it is not clear how the dose of honey to be administered was decided on. This should be clarified, as proper conclusion can not be reached if it is not known whether there may be a more optimal dose of honey administration.

With regards to the antioxidant and oxidative stress testing, it appears that the authors did tests that they had available, rather than strategically deciding which tests would yield the appropriate data. GPX activity of erythrocytes is good to measure alongside GSH but it would have been better to test GSH (reduced glutathione) and GSSG (oxidised glutathione) and to calculate the ratio. This is probably the best marker of systemic redox balbance. With only GSH only you would also need to do total glutathione and subtract the GSH to find a calculated GSSG. GPX activity, as with all antioxidant enzyme activity tests may be debated as a reliable marker in that once removed from the body the enzyme is now tested in a different environment and thus you are testing the potential of the enzyme, rather than how well it was working 'in vivo'. MDA tests alone are also not considdered to be proof enough of lipid peroxydation. This is usually paired with another test, such as conjugated dienes, which is an earlier (and more sensative) marker of lipid peroxidation. The use of FRAP may be considdered as a measure of antioxidant capacity, but it may be motivated why this was chosen over other methods such as ORAC or TEAC for example. All of these have some strengths and weaknesses. I would thus not say your tests are not useable, but motivations should be made in some cases and in others it should be considdered when discussed that this is not optimal tests.

With regard to the statistical analysis, it i not clear whether a post hoc test was used for the one way ANOVA, which may be recommended. It is also recommended that a two way ANOVA be considered, as there is more than one intervention in one group. This will show whether there is synergy between insulin and honey treatment or not.

Beyond this it appears that the study design is sound and that the results yielded do indicate that Honey may have some role in the management of diabetes, with some positive outcomes regarding the sensory nerve conduction. I believe with some of the above issues addressed this can be a fantastic manuscript.

6. PLOS authors have the option to publish the peer review history of their article (what does this mean?). If published, this will include your full peer review and any attached files.

Reviewer #1: No

Reviewer #2: No

Reviewer #3: No

---

## [Author Response · Author response to Decision Letter 0]

14 Sep 2020

Response to Editor and Reviewer comments:

The rationale for using honey - why?

Diabetic patients tend to crave for sweet tasting foods as it is usually advised to reduce or omit sugar from their diet. Honey has anti-hyperglycemic and anti-inflammatory effect and it is a potent anti-oxidant. Honey being a sweet tasting substance, can be a good supplementation to give the satisfaction and at the same time help to reduce the complications of diabetes.

The justification for dosage of honey employed?

Honey was administered at a dose of 0.5 g/kg body weight once daily for 6 weeks. Though different doses of honey were used in previous studies, most of the reports suggested that lower dose of honey was more effective compared to higher dose. Since the present study was conducted in diabetic rats, reduction in hyperglycemia was considered as the primary criteria. Nazir L. et al., administered honey to type 2 diabetic subjects postprandial and compared the effects of 75 gm honey with 30 gm honey and observed that 30 gm honey consumption showed more reduction in hyperglycemia than 75 gm (1). The better effect was shown with 30 gm honey, which was 0.5 gm of honey per kg body weight, taking the average human body weight as 60 kg. In previous animal studies, different doses of honey were found to be effective. Erejuwa et al., reported that 1 gm honey per kg body weight was effective for the hypoglycemic and antioxidant effect (2). However, Azman K F et al., showed that 0.2 gm of honey per kg body weight had the neuroprotective effect (3). In both studies, honey was given for a duration of 4 weeks. Taking both reports into consideration and also the results of human study by Nazir et al, we decided to administer 0.5 gm of honey per kg body weight. 

1. Nazir L, Samad F, Haroon W, Kidwai SS, Siddiqi S, Zehravi M. Comparison of glycaemic response to honey and glucose in type 2 diabetes. J Pak Med Assoc. 2014 Jan;64(1):69–71.

2. Omotayo EO, Gurtu S, Sulaiman SA, Ab Wahab MS, K.N.S S, Salleh MSM. Hypoglycemic and Antioxidant Effects of Honey Supplementation in Streptozotocin-induced Diabetic Rats. Int J Vitam Nutr Res. 2010; 80: 74–82.

3. Azman KF, Zakaria R, Othman Z, Aziz CBA. Neuroprotective effects of Tualang honey against oxidative stress and memory decline in young and aged rats exposed to noise stress. Journal of Taibah University for Science. 2018 May 4;12(3):273–84.

The model of diabetic neuropathy - how do you prove and validate this?

In the present study we have selected a model of type 2 diabetic neuropathy that mimics the major physiologic and metabolic changes found in humans. Though, streptozotocin (STZ) and alloxan induced models are the most commonly used methods for induction of diabetes, these models mimic changes closer to type 1 diabetes by causing marked destruction of the pancreatic cell mass rather than type 2 diabetes. Hence, we used diet+STZ induced diabetic neuropathy (DN) model in the present study to mimic the physiologic and pathologic changes of type 2 diabetes in humans. In recent studies, usage of high fat diet + low dose of STZ model was very common to induce type 2 diabetic neuropathy (1-3). However, the composition of diet and dose of streptozotocin varied according to the study.

Ingestion of high-fat diet (HFD) and high-sugar diet for 8 weeks followed by single dose injection of STZ at a dose of 35 mg, i.p., as described by Dang JK et al., was the procedure followed in the present study for inducing type 2 diabetic neuropathy in Wistar rats (1). In our study, after 8 weeks of high fat and high sugar diet, experimental group rats were injected with streptozotocin (intraperitoneally) at a dose of 35 mg/kg body weight, dissolved in citrate buffer. Three days after STZ injection, blood glucose levels of rats were measured from fasting samples. Rats with blood glucose concentration of 200 mg/dl or more were considered diabetic. After the development of diabetes, rats were allowed four more weeks to remain diabetic till they developed neuropathy. Every second week nerve conduction study, a standard electrophysiological investigation was done to confirm the development of neuropathy. All the diabetic rats of experimental group developed neuropathy after 4 weeks of developing diabetes (at 12th week of study), which was confirmed by significant reduction in conduction velocity of sensory and motor nerve when compared to the value of Non-diabetic control group [Data of FBG and NCS at 12th week has been included in the manuscript (Table-1)]. 

1. Jiang-Kun Dang, Yan Wu, Hong Cao, Bo Meng, Cong-Cong Huang,Guo Chen et al. Establishment of a Rat Model of Type II Diabetic Neuropathic Pain. Pain Medicine 2014; 15: 637–64

2. David André Barrière, Christophe Noll, Geneviève Roussy, Farah Lizotte, Anissa Kessai, Karyn Kirby et al. Combination of high-fat/highfructose diet and low-dosestreptozotocin to model long-term type-2 diabetes complications. Scientific Reports. 2018. 8:424-9

3. Jayshree Shriram Dawane, Vijaya Anil Pandit, Madhura Shirish Kumar Bhosale, Pallawi Shashank Khatavkar. Evaluation of Effect of Nishamalaki on STZ and HFHF Diet Induced Diabetic Neuropathy in Wistar Rats. Journal of Clinical and Diagnostic Research. 2016.10(10): 01-05

Several methodological issues that need to be addressed.

Numerous grammatical and formatting errors that should be corrected. 

Response to the Review Comments:

Reviewer #1: 

1. First sentence of the abstract “Diabetic neuropathy is the most common and debilitating complication affecting more than 50% of patients with longstanding diabetes.” I am not sure the word “debilitating” is appropriated.

Modified accordingly – lane 52

2. Authors wrote “Diabetic neuropathy was induced with high fat and high sugar diet for 8 weeks followed by low dose of streptozotocin”. How can Authors be sure rats developed diabetic neuropathy? What was the criteria? Diabetic status is defined according to fasting hyperglycemia; how specific diabetic neuropathy assessed here?

Ingestion of high-fat diet (HFD) and high-sugar diet for eight weeks followed by single dose injection of STZ at a dose of 35 mg, i.p., the procedure as described by Dang JK et al., was followed in the present study, for inducing type 2 diabetic neuropathy in Wistar rats (1). According to Dang et al, after the development of diabetes, rats developed neuropathy by the end of 2 weeks. Dang et al., confirmed neuropathy by Mechanical Withdrawal Threshold and Thermal Withdrawal Latency. We have done nerve conduction study which is a standard electrophysiological investigation to confirm the development of neuropathy at the end of second week and fourth week after the development of diabetes. After 4 weeks of developing diabetes (at 12th week of study), the rats of experimental group developed neuropathy, which was confirmed by significant reduction in conduction velocity of sensory and motor nerve when compared to the value of Non-diabetic control group [Data of FBG and NCS at 12th week is included in the manuscript(Table-1)]. 

1. Jiang-Kun Dang, Yan Wu, Hong Cao, Bo Meng, Cong-Cong Huang,Guo Chen et al. Establishment of a Rat Model of Type II Diabetic Neuropathic Pain. Pain Medicine 2014; 15: 637–64

3. Lane 64 “Also, malondialdehyde an oxidant” style must be improved.

Modified accordingly - Lane 72

4. Lane 69 “improve d”

Corrected - lane 76

5. Lane 146 “dose of 0.5 gm/kg body”

 Modified accordingly - lane 155

6. Lane 146 no information is provided concerning administration frequency of honey.

Added: Honey was administered once daily at a dose of 0.5 gm/kg body weight using an oral cannula. Honey was diluted with distilled water and the dilution was prepared freshly each time it was administered – lane 154 to 156

7. Lane 154 “None of the humane end points”

Modified accordingly – lane 163 to 164

8. Lane 160 “Two ml of blood sample was collected”

Modified accordingly - lane 170

9. Lane 162 Authors wrote “the neuropathy marker neuron specific enolase (NSE).” I am not sure NSE is specific of neuropathology. Authors should develop on this.

The lane (172) was modified to neuron specific enolase (NSE). However, we submit here reports that state that NSE is specific to neuropathology.

1. Haque A, Polcyn R, Matzelle D, Banik NL. New Insights into the Role of Neuron-Specific Enolase in Neuro-Inflammation, Neurodegeneration, and Neuroprotection. Brain Sci. 2018 Feb 18;8(2). Available from: https://www.ncbi.nlm.nih.gov/pmc/articles/PMC5836052/

2. M. A, Ummer V S, Maiya AG, Hande M, V.s. B. Effect of photobiomodulation on serum neuron specific enolase (NSE) among patients with diabetic peripheral neuropathy-A pilot study. Diabetes & Metabolic Syndrome: Clinical Research & Reviews. 2020;14(5):1061-3.

3. Polcyn R, Capone M, Hossain A, Matzelle D, Banik NL, Haque A. Neuron specific enolase is a potential target for regulating neuronal cell survival and death: implications in neurodegeneration and regeneration. Neuroimmunol Neuroinflamm. 2017;4:254–7.

10. Lane 175. Authors wrote “Thiobarbituric acid (TBA) was measured by the method of Yagi et al. [21] and expressed in terms of malondialdehyde (MDA).” Actually, TBA test evaluates substances that can react with thiobarbituric acids (TBARS for thiobarbituric acid reactive subtances). Then it can certainly not just be “expressed in terms of MDA”.

Modified accordingly - lane 186 to 190

11. Lane 176 Authors wrote “Total antioxidant status (TAS) was measured by ferrous-reducing antioxidant power [FRAP] assay”. I am not sure FRAP is sufficient to evaluate TAS.

We submit here reports that state FRAP assay is a test to assess total antioxidant power. 

1. Benzie IFF, Strain JJ. [2] Ferric reducing/antioxidant power assay: Direct measure of total antioxidant activity of biological fluids and modified version for simultaneous measurement of total antioxidant power and ascorbic acid concentration. In: Methods in Enzymology. Academic Press; 1999. p. 15–27. (Oxidants and Antioxidants Part A; vol. 299). Available from: http://www.sciencedirect.com/science/article/pii/S0076687999990055

2. Benzie IFF, Szeto YT. Total Antioxidant Capacity of Teas by the Ferric Reducing/Antioxidant Power Assay. J Agric Food Chem. 1999 Feb 1;47(2):633–6.

3. Griffin SP, Bhagooli R. Measuring antioxidant potential in corals using the FRAP assay. Journal of Experimental Marine Biology and Ecology. 2004 May 12;302(2):201–11.

12. Table 1. Three significant numbers are sufficient.

We consulted the biostatistician of our institute regarding the presentation of style of referring the level of significance in the table and we were advised to keep the present style of presentation of more than three significant numbers in the table. To make the readers comprehend the differences of results of the groups. Hence, we request our esteemed reviewer to kindly accept the present format of statistical significance in the tables. 

13. Lane 242 “the percentage change in HDL and AIP was significantly more in both …” A word seems missing in the sentence.

Modified accordingly - lane 274

14. Figures 2 and 3 bring duplicate information of table 1.

In Figures 2 and 3, the percentage changes from baseline to the 6th week values have been depicted, where as in Table -1 (present Table 2), the differences in absolute values between the groups post six weeks of intervention have been depicted.

15. Lane 253 “MDA, GPx and NSE were significantly more, whereas…” text should be more specific, more intense? More impacted…? Same remarks for lane 256

Modified accordingly – lane 285 to 290

16. Figures 4 and 5 bring duplicate information of table 2.

In Figures 4 and 5, the percentage changes from baseline to the 6th week values have been depicted, where as in Table -2 (present Table 3), the differences in absolute values between the groups post six weeks of intervention have been depicted.

17. Same remark for table 2: please note three significant numbers are sufficient. Then table message is more readable than when containing too may numbers.

We consulted the biostatistician of our institute regarding the presentation of style of referring the level of significance in the table and we were advised to keep the present style of presentation of more than three significant numbers in the table. To make the readers comprehend the differences of results of the groups. Hence, we request our esteemed reviewer to kindly accept the present format of statistical significance in the tables. 

18. Lane 289 “significant re duction”

Corrected lane 321

19. How come a r coefficient be superior to 1? For TG, r was equal to 6.6481! Again, three significant numbers are sufficient. I think this table should be removed. 

As suggested by the esteemed reviewer, we removed this table 

20. Figure 6. there is a typo “advance glycation end products”. I do not understand why authors put such a schematic scheme. It proposes no hypothetical explanation of honey treatment in their experimental model of DN.

As per the inputs of revered reviewer, we removed figure - 6

21. Authors should provide a table with information about regimen in term of composition for standard rodent chow (SRC) and SRC plus honey.

SRC Composition

Composition of ERC Percentage

Crude Protein 18.54

Crude Fat 3.34

Carbohydrates 65.00

Crude Fibre 5.50

Calcium 1.22

Phosphorous 0.58

Total Ash 5.65

Moisture 7.72

Energy 3085 kcal/kg

Honey was administered diluted in distilled water using an oral cannula once daily for 6 weeks. The dilution was prepared freshly each time it was administered.

22. Authors should consider adding a “limitations” section in their discussion.

Added - lane 438

Reviewer #2: 

1. The aim of the study was to explain scientific basis of adjunct therapy in the treatment of diabetic neuropathy, in this case, honey. Can the authors perhaps include a statement justifying inclusion of insulin as well? And still on that aspect, their model of diabetes is type 2, so why not go for drugs such as metformin or any other drugs commonly used in the treatment of type 2 diabetes?

There are reports on the effect of honey when compared to the anti-diabetic drugs such as metformin and glibenclamide (1-2). These studies have reported that when honey was given with metformin or glibenclamide, there was reduction in hyperglycemia and oxidative stress when compared to metformin or glibenclamide alone. Since there are established reports on these drugs, we have selected insulin to compare with honey. Also, we have used streptozotocin to develop diabetes in our study. As streptozotocin causes β- cell destruction, we used insulin treatment modality in the present study.

1. Erejuwa OO, Sulaiman SA, Ab Wahab MS et al. Glibenclamide or Metformin Combined with Honey Improves Glycemic Control in Streptozotocin-Induced Diabetic Rats. Int J Biol Sci 2011; 7: 244–252.

2. Ozra N, Reza H, Fatima R, Farah F. Effect of natural honey from Ilam and metformin for improving glycemic control in streptozotocin-induced diabetic rats. Avicenna J Phytomedicine. 2012; 2: 212-221.

3. Erejuwa OO, Sulaiman SA, Wahab MS, et al. Antioxidant protective effect of glibenclamide and metformin in combination with honey in pancreas of streptozotocin-induced diabetic rats. Int J Mol Sci. 2010; 11: 2056-66. 

4. Erejuwa OO, Sulaiman SA, Wahab MS, et al. Comparison of antioxidant effects of honey, glibenclamide, metformin, and their combinations in the kidneys of streptozotocin-induced diabetic rats. Int J Mol Sci. 2011; 12: 829-43

2. In the abstract specify the STZ dose used instead of only saying “low dose of STZ” and the type of diabetes as well also in the materials and methods sections as the type only appears in the discussion section.

Modified accordingly- lane 59 to 61

3. The methodology in the abstract omits the 4 week period where the rats were on the standard diet with the nerve conduction study being carried out. It also does not mention how neuropathy was confirmed.

Modified accordingly - lane 61 to 67

4. In table 1, please include the baseline weights or measured before treatment to give a clearer picture to the significant weight differences.

Baseline data of body weight, FBG and NCS are depicted in Table 1.

5. I find some sentences in the section on “Effect of six week intervention on lipid Profile and lipid risk factors” confusing. For example, the sentence in lines 231 to 232….”Lipid profile and lipid risk factors were significantly high and HDL-C was significantly less…..” Isn’t HDL-C part of the lipid profile? Perhaps using conjunction such as: “with the exception of” will help with this.

Modified accordingly - lane 263 to 267

6. Also paraphrase the statement in lines 400 to 402 as it stands it is confusing.

Modified accordingly - lane 418 to 420

7. Please paraphrase for clarity the sentence in lines 288 to 291…”Comparison of NCS…..in DN+insulin group”

Modified accordingly - lane 320 to 323

8. Can the authors include a reference to the point in lines 426- 427…”almost close to four years of treatment of honey in human beings.”

Added references for rat and human age comparison: (lane 449)

1. Sengupta P. The Laboratory Rat: Relating Its Age With Human’s. Int J Prev Med. 2013 Jun;4(6):624–30.

2. Andrello NA, Santos EFD, Araujo MR, Lopes LR. Rat’s age versus human’s age: what is the relationship? ABCD Arq Bras Cir Dig 2012;25(1):49-51

9. Minor issue Line 399 though instead of through

Corrected- lane 417

Reviewer #3: 

Comment no.1: The study appears to be controlled sufficiently through the positive and negative controls employed in this study. It is a pity that some rats died unexpectedly in thus study. It may be advisable to continually measure glucose levels in future studies of this nature, as this is not unheard of in STZ induced diabetic models.

Reply to Comment No.1:

We sincerely thank the esteemed reviewer for the suggestion. In our future studies, we will do as per your advice and measure glucose values every day. However, kindly refer to the following articles that states about the mortality rate of STZ induced diabetes. 

1. Jiang-Kun Dang, Yan Wu, Hong Cao et al. Establishment of a Rat Model of Type II Diabetic Neuropathic Pain. Pain Medicine 2014;15: 637–664

2. Wang Y-J, Xie X-S, Feng S-G, Long Q-X, Ai N, Wang B-F. [Causes of death in STZ-induced rat models of diabetes mellitus]. Sichuan Da Xue Xue Bao Yi Xue Ban. 2014 Jul;45(4):691–5.

3. Deeds MC, Anderson JM, Armstrong AS, Gastineau DA, Hiddinga HJ, Jahangir A, et al. Single dose streptozotocin-induced diabetes: considerations for study design in islet transplantation models. Lab Anim. 2011 Jul 1;45(3):131–40.

4. Wang-Fischer Y, Garyantes T. Improving the Reliability and Utility of Streptozotocin-Induced Rat Diabetic Model [Internet]. Journal of Diabetes Research. 2018 [cited 2020 Sep 7]. Available from: https://www.hindawi.com/journals/jdr/2018/8054073

It is to be noted that the mortality rate of experimental rats in Streptozotocin induced diabetes was more in other studies, which is comparatively less in our present study.

Comment no.2: One criticism of the intervention would be that it is not clear how the dose of honey to be administered was decided on. This should be clarified, as proper conclusion cannot be reached if it is not known whether there may be a more optimal dose of honey administration.

Reply to Comment No.2:

Honey was administered at a dose of 0.5 g/kg body weight once daily for 6 weeks. Though different doses of honey were used in previous studies, most of the reports suggested that lower dose of honey was more effective compared to higher dose. Since the present study was conducted in diabetic rats, reduction in hyperglycemia was considered as the primary criteria. Nazir L. et al., administered honey to type 2 diabetic subjects postprandial and compared the effects of 75 gm honey with 30 gm honey and observed that 30 gm honey consumption showed more reduction in hyperglycemia than 75 gm (1). The better effect was shown with 30 gm honey, which was 0.5 gm of honey per kg body weight, taking the average human body weight as 60 kg. In previous animal studies, different doses of honey were found to be effective. Erejuwa et al., reported that 1 gm honey per kg body weight was more effective for the hypoglycemic and antioxidant effect (2). However, Azman K F et al., showed that 0.2 gm of honey per kg body weight had the neuroprotective effect (3). In both studies, honey was given for a duration of 4 weeks. Taking both reports into consideration and also the results of human study by Nazir et al, we decided to administer 0.5 gm of honey per kg body weight. 

1. Nazir L, Samad F, Haroon W, Kidwai SS, Siddiqi S, Zehravi M. Comparison of glycaemic response to honey and glucose in type 2 diabetes. J Pak Med Assoc. 2014 Jan;64(1):69–71.

2. Omotayo EO, Gurtu S, Sulaiman SA, Ab Wahab MS, K.N.S S, Salleh MSM. Hypoglycemic and Antioxidant Effects of Honey Supplementation in Streptozotocin-induced Diabetic Rats. Int J Vitam Nutr Res. 2010; 80: 74–82.

3. Azman KF, Zakaria R, Othman Z, Aziz CBA. Neuroprotective effects of Tualang honey against oxidative stress and memory decline in young and aged rats exposed to noise stress. Journal of Taibah University for Science. 2018 May 4;12(3):273–84.

Comment no.3: With regards to the antioxidant and oxidative stress testing, it appears that the authors did tests that they had available, rather than strategically deciding which tests would yield the appropriate data. GPX activity of erythrocytes is good to measure alongside GSH but it would have been better to test GSH (reduced glutathione) and GSSG (oxidised glutathione) and to calculate the ratio. This is probably the best marker of systemic redox balance. With only GSH only you would also need to do total glutathione and subtract the GSH to find a calculated GSSG. GPX activity, as with all antioxidant enzyme activity tests may be debated as a reliable marker in that once removed from the body the enzyme is now tested in a different environment and thus you are testing the potential of the enzyme, rather than how well it was working 'in vivo'. 

Reply to Comment No.3: We agree with the inputs of the learned reviewer statement that GSH:GSSG ratio is best indicator of oxidative stress. We could not measure GSSG due to the practical difficulties of non-availability of the measurement method in our laboratory. We agree that this is one of our limitation in the estimation of oxidative stress in the present study. 

Comment no.4: MDA tests alone are also not considered to be proof enough of lipid peroxidation. This is usually paired with another test, such as conjugated dienes, which is an earlier (and more sensitive) marker of lipid peroxidation. 

Reply to Comment No.4: We agree with the comments of the esteemed reviewer that lipid peroxidation can be estimated better when MDA analysis is paired with conjugated dienes. Estimation of conjugated dienes is expensive and time consuming. Moreover, there are number of studies which states that MDA is the best marker of lipid peroxidation (1-3). Taking these reports and difficulty in the estimation of conjugated dienes into consideration we decided to do MDA assay alone for identifying lipid peroxidation in the study. 

1. Gaweł S, Wardas M, Niedworok E, Wardas P. [Malondialdehyde (MDA) as a lipid peroxidation marker]. Wiad Lek. 2004;57(9–10):453–5.

2. Tsikas D. Assessment of lipid peroxidation by measuring malondialdehyde (MDA) and relatives in biological samples: Analytical and biological challenges. Analytical Biochemistry. 2017 May 1;524:13–30.

3. Grotto D, Maria LS, Valentini J, Paniz C, Schmitt G, Garcia SC, et al. Importance of the lipid peroxidation biomarkers and methodological aspects FOR malondialdehyde quantification. Química Nova. 2009;32(1):169–74.

Comment no.5: The use of FRAP may be considered as a measure of antioxidant capacity, but it may be motivated why this was chosen over other methods such as ORAC or TEAC for example. All of these have some strengths and weaknesses. I would thus not say your tests are not useable, but motivations should be made in some cases and in others it should be considered when discussed that this is not optimal tests.

Reply to Comment No.5: We agree with the inputs of the learned reviewer that Oxygen Radical Absorbance Capacity (ORAC) and Trolox Equivalent Antioxidant Capacity (TEAC) are better tests for estimating total antioxidant capacity. ORAC is the most widely recognized method of all the antioxidant assays, the reaction is simple in concept but it is more complex in practice. As there are some practical issues in performing ORAC, we could not use this test in our study for estimating total anti-antioxidant capacity. 

a. Careful control and monitoring of ORAC reaction temperature is needed. When required temperatures are not reached, reactions are slow and incomplete, and results are poorly reproducible. Sometimes, slow reaction can be misinterpreted as increased radical scavenging, leading to an overestimation of antioxidant activity. Reaction temperature cannot be achieved by setting on an instrument. The most accurate control is obtained by reactions in single cells in heated holding blocks and fluorescence sample blocks but this eliminates rapid output. Obtaining required temperatures with plate readers is also difficult.

b. Careful control of oxygen is critical. Full and reproducible oxygenation is required for the azide reaction to efficiently generate radicals. However, solubility of oxygen declines as temperature increases, so oxygen becomes depleted when solutions are pre-warmed before runs, during thermal equilibration in ovens, and when reaction must be heated over long times due to strong antioxidant inhibition. Variations in handling and heating times between samples cause considerable variability in dissolved pO2, hence inconsistent results. With insufficient oxygen, reactions are slow, variable, and do not run to completion. In our laboratory setup, maintaining the oxygen control was difficult. 

 In TEAC, the reaction takes place at a faster rate. Due to this accurate results can only be obtained with rapid mixing methods such as autodispensing plate readers or stopped-flow mixers, which made the assay out of our routine laboratory capabilities. 

Reports suggests that FRAP assay is comparable with both of these methods (1,2) and used in various studies for measuring the total antioxidant capacity. The FRAP assay is simple and also feasible for us to do in our laboratory setup. Reagents of FRAP are inexpensive, of low toxicity and high stability. The procedure of FRAP is technically straightforward and the results are available within a few minutes of reagent/sample mixing (4min at the recommended reaction temperature to 37°C). Also, FRAP assay has high sensitivity and precision. Taking these points into consideration, we selected FRAP assay in our study for the estimation of total antioxidant capacity.

1. Rao PVLNS, V.S.Kiranmayi, P.Swathi, Jeyseelan L, M.M.Suchitra, Bitla AR. Comparison of Two Analytical Methods Used For the Measurement of Total Antioxidant Status. JAA. 2015 Jun 6;1(1):22.

2. Dudonné S, Vitrac X, Coutière P, Woillez M, Mérillon J-M. Comparative Study of Antioxidant Properties and Total Phenolic Content of 30 Plant Extracts of Industrial Interest Using DPPH, ABTS, FRAP, SOD, and ORAC Assays. J Agric Food Chem. 2009 Mar 11;57(5):1768–74.

Comment no.6: With regard to the statistical analysis, it is not clear whether a post hoc test was used for the one way ANOVA, which may be recommended. It is also recommended that a two way ANOVA be considered, as there is more than one intervention in one group. This will show whether there is synergy between insulin and honey treatment or not.

Reply to Comment No.6: We have used Bonferroni post-hoc test for the one way ANOVA analysis (included in Manuscript lane 213). We are grateful to the suggestion of our revered reviewer. As the primary objective of the study was to see the effect of honey on diabetic neuropathy complications, we did not do the two-way ANOVA to see the synergy between honey and insulin treatment.

---

## [Decision Letter · Decision Letter 1]

28 Oct 2020

PONE-D-20-18538R1

Effect of honey and insulin treatment on oxidative stress and nerve conduction in an experimental model of diabetic neuropathy Wistar rats

PLOS ONE

Dear Dr. Gaur,

Thank you for submitting your manuscript to PLOS ONE. After careful consideration, we feel that it has merit but does not fully meet PLOS ONE’s publication criteria as it currently stands. Therefore, we invite you to submit a revised version of the manuscript that addresses the points raised during the review process.

Two of the reviewers raised concerns regarding your revised manuscript. Please address all these queries. Also, make sure all your edits (from the previous revision and the current one) are incorporated in the MS and not only when you address the reviewer, i.e. response to reviewers. This point was made strongly, and correctly so, by Reviewer #3 and must be addressed for all corrections/responses made thus far and corrections still to be made, and must reflect in the revised MS itself.  Please note that it is essential to ensure all corrections are satisfactorily made and as requested by the reviewers.

We look forward to receiving your revised manuscript.

Kind regards,

M. Faadiel Essop

Academic Editor

PLOS ONE

Reviewers' comments:

Reviewer's Responses to Questions

**Comments to the Author**

1. If the authors have adequately addressed your comments raised in a previous round of review and you feel that this manuscript is now acceptable for publication, you may indicate that here to bypass the “Comments to the Author” section, enter your conflict of interest statement in the “Confidential to Editor” section, and submit your "Accept" recommendation.

Reviewer #1: (No Response)

Reviewer #2: (No Response)

Reviewer #3: All comments have been addressed

2. Is the manuscript technically sound, and do the data support the conclusions?

Reviewer #1: Yes

Reviewer #2: Yes

Reviewer #3: Yes

3. Has the statistical analysis been performed appropriately and rigorously? 

Reviewer #1: (No Response)

Reviewer #2: Yes

Reviewer #3: Yes

4. Have the authors made all data underlying the findings in their manuscript fully available?

Reviewer #1: Yes

Reviewer #2: Yes

Reviewer #3: Yes

5. Is the manuscript presented in an intelligible fashion and written in standard English?

Reviewer #1: Yes

Reviewer #2: Yes

Reviewer #3: Yes

6. Review Comments to the Author

Reviewer #1: In their revised version of their MS, Authors correctly addressed most of points I raised.

Still, I do have several questions:

Concerning point #11. Lane 176 Authors wrote “Total antioxidant status (TAS) was measured by ferrousreducing antioxidant power [FRAP] assay”. I am not sure FRAP is sufficient to evaluate

TAS.

In your answer you submit reports that state FRAP assay is a test to assess total antioxidant

power.

Actually my point is FRAP, despite being a very good test to assess antioxidant capacity of a solution/molecule, is not sufficient to evaluate TAS. Additional tests should be used to be sure to evaluate TAS such as ORAC, DDPH…

Concerning point #21. Authors should provide a table with information about regimen in term of

composition for standard rodent chow (SRC) and SRC plus honey.

Authors give data in their answer but with no unit. The table with regimen composition and correct units should be included in the revised version of the MS.

There are still typos in the MS, Authors should proofread carefully before resubmitting.

Lane 420 authors wrote “hypolipedemic effect was moderate”

Reviewer #2: The authors have addressed fully comments from reviewer 1 and 2 but not entirely those from reviewer 3.

Reviewer #3: The reviewer and editor comments appear to have been addressed with care and I applaud your level of honesty in admitting when you simply can not afford or accomplish specific methods. My concern is that beyond defending your manuscript to the editor and reviewers you made very little changes to the manuscript itself. I believe the comments are made to improve the quality of the manuscript and some of the motivations made need to be reflected in the published work. As such I believe the answers given satisfactory, but would like to see it reflected in the manuscript.

7. PLOS authors have the option to publish the peer review history of their article (what does this mean?). If published, this will include your full peer review and any attached files.

Reviewer #1: No

Reviewer #2: No

Reviewer #3: No

---

## [Author Response · Author response to Decision Letter 1]

11 Nov 2020

PONE-D-20-18538R1

Effect of honey and insulin treatment on oxidative stress and nerve conduction in an experimental model of diabetic neuropathy Wistar rats

 Review Comments to the Author

Reviewer #1: In their revised version of their MS, Authors correctly addressed most of points I raised. Still, I do have several questions:

Concerning point #11. Lane 176 Authors wrote “Total antioxidant status (TAS) was measured by ferrous reducing antioxidant power [FRAP] assay”. I am not sure FRAP is sufficient to evaluate TAS. In your answer you submit reports that state FRAP assay is a test to assess total antioxidant power. Actually my point is FRAP, despite being a very good test to assess antioxidant capacity of a solution/molecule, is not sufficient to evaluate TAS. Additional tests should be used to be sure to evaluate TAS such as ORAC, DDPH…

Answer: We agree with the inputs of the learned reviewer that Oxygen Radical Absorbance Capacity (ORAC) and Diphenyl-β-picrylhydrazyl (DPPH) are better tests for estimating total antioxidant capacity. ORAC is the most widely recognized method of all the antioxidant assays, the reaction is simple in concept but it is more complex in practice. As there are some practical issues in performing ORAC such as careful control and monitoring of ORAC reaction temperature and careful control of oxygen are critical, we could not use this test in our study for estimating TAS. 

In DPPH, the reaction needs longer duration. The method is based on the spectrophotometric measurement of DPPH concentration changes resulting from the DPPH reaction with an antioxidant. The amount of remaining DPPH in the examined system is a measure of the antioxidant activity of compounds. The study shows that the type and amount of solvent used for the antioxidant dissolution, water content, and hydrogen or metal ion concentration in the measuring system significantly influences the differences in the amount of unreacted DPPH. Due to this presented relationships, standardising the DPPH method shows the complexity of the problem, even in very simple DPPH/antioxidant systems, which is one of the reason for not selecting DPPH for measuring TAS. 

However, reports suggests that FRAP assay is comparable with both of these methods (1,2) and used in various studies for measuring the TAS. The FRAP assay is simple and also feasible for us to do in our laboratory setup. Reagents of FRAP are inexpensive, of low toxicity and high stability. The procedure of FRAP is technically straightforward and the results are available within a few minutes of reagent/sample mixing (4min at the recommended reaction temperature to 37°C). Also, FRAP assay has high sensitivity and precision. Taking these points into consideration, we selected FRAP assay in our study for the estimation of TAS. 

References: 

1. Rao PVLNS, V.S.Kiranmayi, P.Swathi, Jeyseelan L, M.M.Suchitra, Bitla AR. Comparison of Two Analytical Methods Used For the Measurement of Total Antioxidant Status. JAA. 2015 Jun 6;1(1):22.

2. Dudonné S, Vitrac X, Coutière P, Woillez M, Mérillon J-M. Comparative Study of Antioxidant Properties and Total Phenolic Content of 30 Plant Extracts of Industrial Interest Using DPPH, ABTS, FRAP, SOD, and ORAC Assays. J Agric Food Chem. 2009 Mar 11;57(5):1768–74.

Concerning point #21. Authors should provide a table with information about regimen in term of composition for standard rodent chow (SRC) and SRC plus honey.

Authors give data in their answer but with no unit. The table with regimen composition and correct units should be included in the revised version of the MS.

There are still typos in the MS, Authors should proofread carefully before resubmitting.

Answer: SRC composition (with units) is included in the revised version of the MS

Lane 420 authors wrote “hypolipedemic effect was moderate”

Answer: Corrected

Reviewer #2: The authors have addressed fully comments from reviewer 1 and 2 but not entirely those from reviewer 3.

Answer: We have included all the corrections/suggestions given by the esteemed reviewers in the new-revised manuscript. 

Reviewer #3: The reviewer and editor comments appear to have been addressed with care and I applaud your level of honesty in admitting when you simply cannot afford or accomplish specific methods. My concern is that beyond defending your manuscript to the editor and reviewers you made very little changes to the manuscript itself. I believe the comments are made to improve the quality of the manuscript and some of the motivations made need to be reflected in the published work. As such I believe the answers given satisfactory, but would like to see it reflected in the manuscript.

Answer: We are grateful to the suggestion of our revered reviewer. As per the suggestion of our esteemed reviewers we have included our responses to the previous review comments in the revised manuscript.

---

## [Decision Letter · Decision Letter 2]

7 Dec 2020

PONE-D-20-18538R2

Effect of honey and insulin treatment on oxidative stress and nerve conduction in an experimental model of diabetic neuropathy Wistar rats

PLOS ONE

Dear Dr. Gaur,

Thank you for submitting your manuscript to PLOS ONE. After careful consideration, we feel that it has merit but does not fully meet PLOS ONE’s publication criteria as it currently stands. Therefore, we invite you to submit a revised version of the manuscript that addresses the points raised during the review process.

Please address the remaining few queries raised by the reviewers.

Please submit your revised manuscript by 31 January 2021. If you will need more time than this to complete your revisions, please reply to this message or contact the journal office at plosone@plos.org. Please include the following items when submitting your revised manuscript:

We look forward to receiving your revised manuscript.

Kind regards,

M. Faadiel Essop

Academic Editor

PLOS ONE

Reviewers' comments:

Reviewer's Responses to Questions

**Comments to the Author**

1. If the authors have adequately addressed your comments raised in a previous round of review and you feel that this manuscript is now acceptable for publication, you may indicate that here to bypass the “Comments to the Author” section, enter your conflict of interest statement in the “Confidential to Editor” section, and submit your "Accept" recommendation.

Reviewer #1: (No Response)

Reviewer #3: All comments have been addressed

2. Is the manuscript technically sound, and do the data support the conclusions?

Reviewer #1: Yes

Reviewer #3: Yes

3. Has the statistical analysis been performed appropriately and rigorously? 

Reviewer #1: I Don't Know

Reviewer #3: Yes

4. Have the authors made all data underlying the findings in their manuscript fully available?

Reviewer #1: Yes

Reviewer #3: Yes

5. Is the manuscript presented in an intelligible fashion and written in standard English?

Reviewer #1: Yes

Reviewer #3: Yes

6. Review Comments to the Author

Reviewer #1: In their revised version of the MS, Authors correctly addressed points I previously raised.

Just few minor points:

- In table 2 please use only 3 significant numbers.

- Same for table 3 and 4

Reviewer #3: Thank you for addressing the review comments. The only shortcoming that I can still point out is the need to show the dietary composition of SRC and Honey. This may be technically difficult, as the honey was dissolved in water. I however, feel it would be a improvement on the current version.

7. PLOS authors have the option to publish the peer review history of their article (what does this mean?). If published, this will include your full peer review and any attached files.

Reviewer #1: No

Reviewer #3: No

---

## [Author Response · Author response to Decision Letter 2]

23 Dec 2020

PONE-D-20-18538R2

Effect of honey and insulin treatment on oxidative stress and nerve conduction in an experimental model of diabetic neuropathy Wistar rats

6. Review Comments to the Author

Reviewer #1: In their revised version of the MS, Authors correctly addressed points I previously raised. Just few minor points:

 In table 2 please use only 3 significant numbers.

- Same for table 3 and 4.

Answer: As per the esteemed reviewer’s suggestion, in table 2, 3 and 4 following decimal, only two significant numbers were given.

If this did not address the query of our esteemed reviewer, we request to provide one example table of the required change. As per the example we will make the changes and submit it as required.

Reviewer #3: Thank you for addressing the review comments. The only shortcoming that I can still point out is the need to show the dietary composition of SRC and Honey. This may be technically difficult, as the honey was dissolved in water. I however, feel it would be a improvement on the current version.

Answer: We thank our learned reviewer for his valuable comments and suggestion. The dietary composition of honey was not specifically estimated by us in the study as that was not our specific objective of the study. However, we asked the company to provide honey analysis report when they provided honey. We are herewith attaching the analysis of honey (FSSAI approved) as supplementary data for your review.

---

## [Editor Report · Decision Letter 3]

30 Dec 2020

Effect of honey and insulin treatment on oxidative stress and nerve conduction in an experimental model of diabetic neuropathy Wistar rats

PONE-D-20-18538R3

Dear Dr. Gaur,

We’re pleased to inform you that your manuscript has been judged scientifically suitable for publication and will be formally accepted for publication once it meets all outstanding technical requirements.

Kind regards,

M. Faadiel Essop

Academic Editor

PLOS ONE
---

## [Editor Report · Acceptance letter]

6 Jan 2021

PONE-D-20-18538R3 

Effect of honey and insulin treatment on oxidative stress and nerve conduction in an experimental model of diabetic neuropathy Wistar rats 

Dear Dr. Gaur:

I'm pleased to inform you that your manuscript has been deemed suitable for publication in PLOS ONE. Congratulations! Your manuscript is now with our production department. 

Kind regards, 

on behalf of

Dr. M. Faadiel Essop 

Academic Editor

PLOS ONE